# Spatial modeling connecting childhood atopic dermatitis prevalence with household exposure to pollutants
Grace Ratley [1,2], Jordan Zeldin[1,2], Ashleigh A. Sun[1], Manoj Yadav[1], Prem Prashant Chaudhary[1] & Ian A. Myles [1] ✉

## Abstract

**Background** Atopic dermatitis (AD) is a chronic, inflammatory disease characterized by dry, pruritic skin. In the U.S., the prevalence of AD has increased over three-fold since the 1970s. We previously reported a geographic association between isocyanate-containing air pollution and AD as well as mechanistic data demonstrating that isocyanates induce skin dysbiosis and activate the host itch receptor TRPA1. However, non-spatial models are susceptible to spatial confounding and may overlook other meaningful associations.

**Methods** We added spatial analysis to our prior model, contrasting pollution data with clinical visits. In addition, we conducted a retrospective case-control survey of childhood exposure to BTEX-related products. Finally, we assessed implicated compounds, in pure form and as part of synthetic fabric, for their effect on the growth and metabolism of skin commensal bacteria.

**Results** Spatial analysis implicate benzene, toluene, ethylbenzene, and, most significantly, xylene (BTEX) compounds. Survey odds ratios for AD were significant for xylene-derived polyester bed sheets (OR = 9.5; CI 2.2–40.1) and diisocyanate-containing wallpaper adhesive (OR = 6.5; CI 1.5–27.8). *Staphylococcus aureus* lives longer on synthetic textiles compared to natural textiles. Meanwhile, synthetic fabric exposure shifts the lipid metabolism of health-associated commensals (*Roseomonas mucosa* and *S. epidermidis*) away from therapeutic pathways.

**Conclusions** We propose that BTEX chemicals in their raw forms and in synthetic products represent a unifying hypothesis for environmentally induced AD flares through their ability to create dysbiosis in the skin microbiota and directly activate TRPA1. Unequal distribution of these pollutants may also influence racial disparities in AD rates.

## Plain language summary

Atopic dermatitis (AD) is a chronic, inflammatory disease characterized by dry, itchy skin that has become increasingly more common since around 1970. We aimed to identify chemicals that may cause atopic dermatitis (eczema). Building on prior work, we discovered that these chemicals could prevent the good bacteria that live on the skin from making the lipids and oils needed to keep human skin healthy. In this study, we combined new research methods with patient surveys. We link eczema to the chemical xylene, which is found in numerous home products. Exposure to xylene, benzene, or isocyanate containing fabrics (polyester, nylon, or spandex) disrupted the normal functions of skin bacteria. Our results indicate exposure to synthetic fabrics and other sources of these chemicals may contribute to eczema and deepen the understanding of how the environment can drive common diseases.

Atopic dermatitis (AD) is a chronic, inflammatory disease characterized by dry, pruritic skin. Descriptions of AD's clinical presentation written 3500 years ago are remarkably similar to those written today[1]. However, the prevalence of AD has increased 3–6-fold since an inflection point circa 1970[2]. There is mounting evidence linking environmental exposures, particularly during early childhood, to the development of AD and other allergic diseases. Today, children immigrating from a developing to an industrialized country before the age of 6 have a higher risk of asthma, allergies, and AD, while those immigrating later in life do not show an

increased risk[3,4]. Taken together, these epidemiologic findings suggest that the biology of AD has remained historically consistent while exposure to AD-inducing chemicals has gone from natural-but-rare to synthetic-and-common. Many associations between AD and patient-reported triggers, such as topical products, wildfires, industrial environments, foods, and clothing have been documented[5–7], but few controlled studies have investigated the mechanism by which these triggers potentiate disease.

Combining a geographic analysis with cell and mouse modeling, we previously demonstrated that (di)isocyanate exposure correlates with

[1]Laboratory of Clinical Immunology and Microbiology, Epithelial Therapeutics Unit, National Institute of Allergy and Infectious Disease, National Institutes of Health, Bethesda, MD, USA. [2]These authors contributed equally: Grace Ratley, Jordan Zeldin. ✉e-mail: mylesi@niaid.nih.gov

clinical visits for AD[6,7]. Isocyanates are components of several known AD-triggering exposures, including wildfires, automobile exhaust, select adhesives, and spandex fabric. Mechanistically, in addition to directly activating the host itch receptor TRPA1, isocyanate compounds counteract the therapeutic benefits of select commensals by inhibiting the production of ceramide-family lipids[6,7]. The neonatal levels of these ceramide-family lipids were recently reported by several groups as the most predictive of subsequent development of AD[8–10].

However, models that account for spatial effects may suggest compounds otherwise overlooked by non-spatial models. Therefore, to assess additional environmental risks for AD we developed a spatial model that corrected for autocorrelation, as measured by Moran's $I$[11]. Although diisocyanates continued to be a significant variable, the spatial model additionally detected xylene compounds as the most significantly associated risk. Clustering pollutants by chemical or industrial similarity implicated several of the so-called BTEX compounds (benzene, toluene, ethylbenzene, and xylene). These aromatic hydrocarbons are primarily used as anti-knock agents in gasoline but are also found in plastics, adhesives, foams, and synthetic textiles.

Going beyond association using in vitro modeling, we demonstrated that pathogenic skin bacteria like *Staphylococcus aureus* live longer on BTEX-synthesized textiles, while beneficial commensals (*S. epidermidis* and *Roseomonas mucosa*) exhibit disrupted metabolism when grown on BTEX-containing fabrics or when directly exposed to BTEX chemicals. Similar to diisocyanates[6,7], xylene directly induces AD-like dermatitis in mice via TRPA1[12,13]. Therefore, we present BTEX-induced dysbiosis and itch activation as a potentially unifying hypothesis for environmental drivers of AD pathogenesis and suggest future mitigation efforts focus on the avoidance of these select chemical agents.

## Methods
### Spatial model of AD and pollutants
The Definitive Health database contains 1.2 billion diagnostic billing codes per year from 2017 to 2019 across 20,000 zip codes in the United States, including Hawaii, Alaska, and the U.S. territories. We signed a paid contract to access their database. The total number of billing codes for all diagnoses, as well as the number of billing codes for AD specifically (L20.9), were collated by zip code. For each disease tested, only providers from relevant medical specialties were selected. For pediatric AD, the specialists were pediatrics, adolescent medicine, and pediatric allergy/immunology.

Facilities that emit pollution from a list of 510 compounds above a prespecified threshold are required to report this information to the Environmental Protection Agency's (EPA's) Toxic Release Inventory. EPA's Risk-Screening Environmental Indicators model the geographic dispersion of air pollution while accounting for wind patterns to estimate the concentration of each pollutant in each census tract of the U.S. in micrograms per cubic centimeter for each year.

Since the zip code centroid represents the location of the provider, rather than the patient, the pollution concentration was aggregated from the surrounding census tracts within a 30-mile radius. These were aggregated with a weighted average, which considered both the population of each census tract and the distance from the zip code centroid, represented by the Eq. 1

$$((N/\sum \iota = 1)\text{Pollutant}_i * \text{Population}_i * N(\text{dist}_i; 0, \sigma))/$$
$$((N/\sum \iota = 1) * \text{Population}_i * N(\text{dist}_i; 0, \sigma)) \quad (1)$$

The distance decay function was a Gaussian curve parameterized based on the average reported driving distance to a primary care doctor of 8.6 miles. For pediatric analysis, the population weights only include people 0-19 years. For some analyses we look at same-year exposure, and for others, we aggregated the prior 5 years of pollution concentration. Tableau Public (Washington, DC) was used for mapping visualization.

We include the following covariates in analysis: the proportion of visits to each specialist, 5-year age brackets as a proportion of the population, deprivation index, population density, and latitude. These covariates undergo the same weighing aggregation scheme as pollution concentration. For the pediatric analysis, only age brackets up to age 19 are included. The deprivation index was taken from the University of Wisconsin's Neighborhood Atlas at the census tract level. Population density was derived by dividing the population in each census tract from the American Community Survey by the surface area of each census tract. Five-year age brackets were divided by the total population at each census tract. To avoid complete multicollinearity, the most common specialist, usually pediatrics or family medicine, as well as the last age bracket are removed as covariates.

A generalized linear model with mixed effects is used to iteratively analyze each pollutant while accounting for covariates and spatial random intercepts, using the lme4 package. Negative binomial and Poisson structures are included with an offset term that represents the number of billable visits. The zip codes are organized into hierarchical clusters based on latitude and longitude. Nested random intercepts are included for clusters of 3, 9, 27, and 81 zip codes. Moran's $I$ is calculated using moran.test from the spdep package, using Pearson residuals and inverse distance decay. Overdispersion was calculated with the overdisp.glmer function in the RVAideMemoire package. $P$ values were adjusted using both the Bonferroni method and Benjamini–Hochberg methods.

To assess chemical structure similarity, we used the publicly available CAS codes to derive CID and canonical SMILES. For chemical mixtures or groups of similar compounds, a representative compound was chosen as follows for RSEI compound grouping and (representative molecule): polychlorinated alkanes (2,4,5,8,11,12,14,17-Octachloroicosane); Glycol ethers (2-Butoxyethanol); Diisocyanates (non-TDI) (3,3'-Dimethyl-4,4'-diphenylene diisocyanate); Polycyclic aromatic hydrocarbons (benzo[a]pyrene); Dioxins (2,3,7,8-Tetrachlorodibenzo-P-dioxin); Nonphenols (4-Nonylphenol); Hexabromocyclododecane (1,2,5,6,9,10-Hexabromocyclododecane); Chlorophenols (2,3,4,6-Tetrachlorophenol).

Extended connectivity fingerprints were determined for each molecule, and the pairwise Tanimoto distance was calculated to create a similarity matrix, which was used to hierarchically cluster the compounds using the complete linkage method. To determine the optimal number of clusters, for each 1 modular 3 clusters, we calculated a weighted mean of intra-cluster dissimilarity: for each cluster, all pairwise Tanimoto distances are averaged together to gauge the dissimilarity of each group; then, all the dissimilarity means for each group are averaged together in a weighed fashion, where the weights are the number of molecules in each group. This index indicates the average dissimilarity in each cluster and monotonically decreases as the number of groups increases. The threshold of 0.85 similarity, considered a benchmark similarity threshold, was used to determine the optimal number of clusters.

### Childhood BTEX exposure survey
The survey was approved by the institutional review board (IRB) of the National Institutes of Health and distributed to the public through Research Match, Global Parents for AD Research, and the Coalition of Skin Diseases. The survey obtained informed consent prior to presenting the included questions. Questions were about participant demographics, AD severity, and childhood exposure to BTEX-related products and processes. Controls were patients with non-AD skin diseases (See supplementary note 1 for all questions).

Sample size was calculated by Eq. 2:

$$n = p(1 - p)(Z/E)^2 \quad (2)$$

Where Z is equal to the value for the standard normal distribution reflecting the confidence level that will be used (Z can be set to 1.96 to obtain our desired confidence interval of 95%). $p$ is equal to the unknown population proportion. Setting p to 0.2 calculated that we needed 246 responses. For participants under the age of 18, the survey was completed

by a parent. Data analysis and visualization were conducted in R, using the packages: pheatmap, ggplot2, epitools, and corrplot. Incomplete survey responses and those who could not recall childhood exposures were not included in the analysis. Percent of body surface area (BSA) affected by AD was calculated using SCORAD. Participants were asked to rate the severity of their itch on an average day on a scale of 1 to 10, as well as the frequency of their itch on a scale of 1–4. A composite metric of itch severity and frequency, giving equal weight to both variables was calculated using Eq. 3:

$$0.5 * (severity/10) + 0.5 * (frequency/4) \qquad (3)$$

### Commensal bacteria growth on textiles

We obtained 5 untreated, white pillowcases with thread counts between 400 and 600 made from common textiles: cotton (Amazon Basics), bamboo (American Home Collection), silk (MAXFEEL), polyester (MEILA), and a mix of 85% nylon and 15% spandex (Deluxe Comfort). *Staphylococcus aureus* isolate USA300 (Sa) and *Staphylococcus epidermidis* (Se) were cultured in BHI broth at 37 °C and *Roseomonas mucosa* (Rm) in R2A broth at 32C[7]. After 48 hours, the broths were diluted to achieve an OD of 0.4. In addition, a 1:1:1 mixture of the broths was prepared. Rough 1-inch squares were cut out of each fabric, autoclaved, and arranged in a single 2 × 2 layer. 1.5 mL of broth was sprayed in the center of the arrangement, ensuring an even distribution of bacteria on each piece. The sheets were allowed to dry, then were incubated at 32 C. At days 3, 10, and 20, a touch plate was made using one of the four fabric squares. For *Staphylococcus* species, tryptic soy agar touch plates were incubated at 37 °C. For Rm and the mixture, R2A agar touch plates were incubated at 32 C. After 48 hours, images were taken using bright field microscopy. Samples of bacteria were collected and stored at −80 °C in 0.5 mL of 100% methanol until mass spectrometry was performed.

### Commensal bacteria exposure to BTEX chemicals

Spiral plates of Rm, Sa, and Se grown for 48 hours under optimal growth conditions and diluted to OD 0.4 were exposed to 10uL of benzene (Sigma), toluene (Sigma), p-xylene (Sigma), 1,2,4-trimethylbenzene (Sigma), and 2,6-dimethylaniline (xylidine; Sigma). Untreated and ethanol-treated plates were used as controls. 24 hours after exposure, cultures were collected in 0.5 mL of 100% methanol and sonicated for 8 pulses with an amplitude of 100um (Q55 Sonicator) in preparation for mass spectrometry.

### Metabolic data collection and untargeted metabolomics analysis

Samples were combined with 2,5-Dihydroxybenzoic acid (DHB) (20 mg/mL; # 149357-10 G Sigma) matrix solution in 70% methanol (Simga), and 0.2% trifluoroacetic acid (TFA; Sigma). 1 µL of the solution was plated on a disposable MTP Target Plate (Bruker; Billercia, MA) in quadruplicate. Matrix-assisted laser desorption ionization, trapped ion mobility spectrometry, time of flight (MALDI-timsTOF) were performed and data was collected using timsControl (Bruker; Billercia, MA) with the following parameters: scan range 150–1300 m/z on positive MS scan mode; TIMS settings of 1/K0, 0.7–1.92 V*s/cm2; ramp time of 200 ms; Accu time of 1209 ms; duty cycle of 100%; and ramp rate of 0.82 Hz. Statistical analyses were conducted in R. Outlier samples were identified on the basis of NMDS values and were removed. Intensity data was pareto scaled and log transformed and unpaired t-test was used to determine differentially abundant peaks. Pathway analysis was conducted in MetaboAnalyst using the mummichog algorithm. The index of pathway significance was calculated from mummichog outputs using the following formula: $[(FET)^2] * (((Significant + 1)^2 + (total-significant))/(pathway×expected))+1$.

### Reporting summary

Further information on research design is available in the Nature Portfolio Reporting Summary linked to this article.

## Results

### Spatial modeling of AD prevalence and pollution reveals a strong association with xylene isomers

Our previous, non-spatial Poisson model of AD prevalence and pollution resulted in a significant Moran's I, indicating the presence of spatial auto-correlation and overdispersion (Fig. 1a; upper panel). Dimensionality reduction was used to improve the model by clustering pollutants by their molecular similarity (Supplemental Fig. 1A, B). The optimization selected a cutoff of 260 molecularly similar clusters (Supplemental Fig. 1C, D). The diisocyanate group continued to be the top association with AD visit rates when clustered by chemistry in non-spatial Poisson models (Fig. 1b). However, Moran's I continued to be significant for this approach ($I = 0.048$; $p < 0.0001$) with overdispersion (ratio ~37). We thus used a negative binomial model with a nested random intercept spatial structure. This model corrected for Moran's I ($I = -0.024$, $p = 1$), visualized by mapping residuals (Fig. 1a; bottom panel) as well as overdispersion (ratio ~1). In this spatial model, 5-year aggregated air pollution exposure among pediatric patients revealed several compounds that were significantly positively associated with AD visit rates (Fig. 1c, d, Supplemental Fig. 2A).

Some associated compounds with unadjusted significance in the new spatial model were ones we previously reported: toluene diisocyanate (mixed isomers), toluene-2,6-diisocyanate, and antimony[6,7,14] (Supplemental Fig. 2A). Several compounds were significant across multiple years or significant after multiple hypothesis testing adjustments (Supplemental Fig. 2A–G). However, only one variable was positively associated with AD after multiple hypothesis correction across all three of our evaluated years: mixed isomers of xylene (Fig. 1e). The model also identified two compounds that were negatively associated with AD visitation rate: silver and thiourea. Xylene pollution shared the most similar spatial distribution with ethyl-benzene and 1,2,4-trimethylbenzene, and shared the most similar molecular structure with these same two compounds (Fig. 1c, Supplemental Fig. 2B–D). Among the BTEX compounds, xylene was associated with AD, food allergy, obesity, and psoriasis but not the other pediatric or atopic diseases (Fig. 1f).

### Survey data suggests BTEX exposure is a risk factor for AD incidence and severity

In a case-control study, participants were asked to recall their childhood exposure (or their child's exposure) to several BTEX-related products and processes. Out of 311 completed surveys, 46% of respondents were female. The mean age was 18 years. 69% of respondents had AD, with the remaining being assigned to the control group (Fig. 2a). There was no evidence of multicollinearity among exposures as measured by the variance inflation factor (VIF < 2.5; Fig. 2b).

There were no differences between patients with AD and controls for the two negative control exposures included in our survey: making changes to outdoor landscaping and replacing kitchen appliances (Fig. 1c). The three highest odds of subsequent AD in respondents were using polyester bed sheets (OR: 9.5; CI: 2.2–40.1), replacing carpeting (OR: 6.7; CI: 2.0–22.1), and adding, replacing, or removing wallpaper (OR: 6.5; CI: 1.5–27.8). In a multivariate model, only polyester was significantly associated with having AD, and denying any of the exposures in our survey was significantly associated with the control group (Fig. 2d). Among patients with AD, denying exposure to BTEX products during childhood had a significant negative correlation with AD severity as measured by BSA (Fig. 2e) and a combined metric accounting for the severity and frequency of reported itch symptoms (Fig. 2f). Meanwhile, using polyester sheets had a significant positive correlation with BSA and itch severity while wallpaper exposure was associated with only itch severity (Fig. 2e, f).

### BTEX-derived fabric exposure influences microbe metabolism and survival

To further elucidate in-home exposures to BTEX, pillowcases made from natural (cotton, bamboo, and silk) and synthetic (nylon/spandex and polyester) textiles, were inoculated with *Roseomonas mucosa* (Rm),

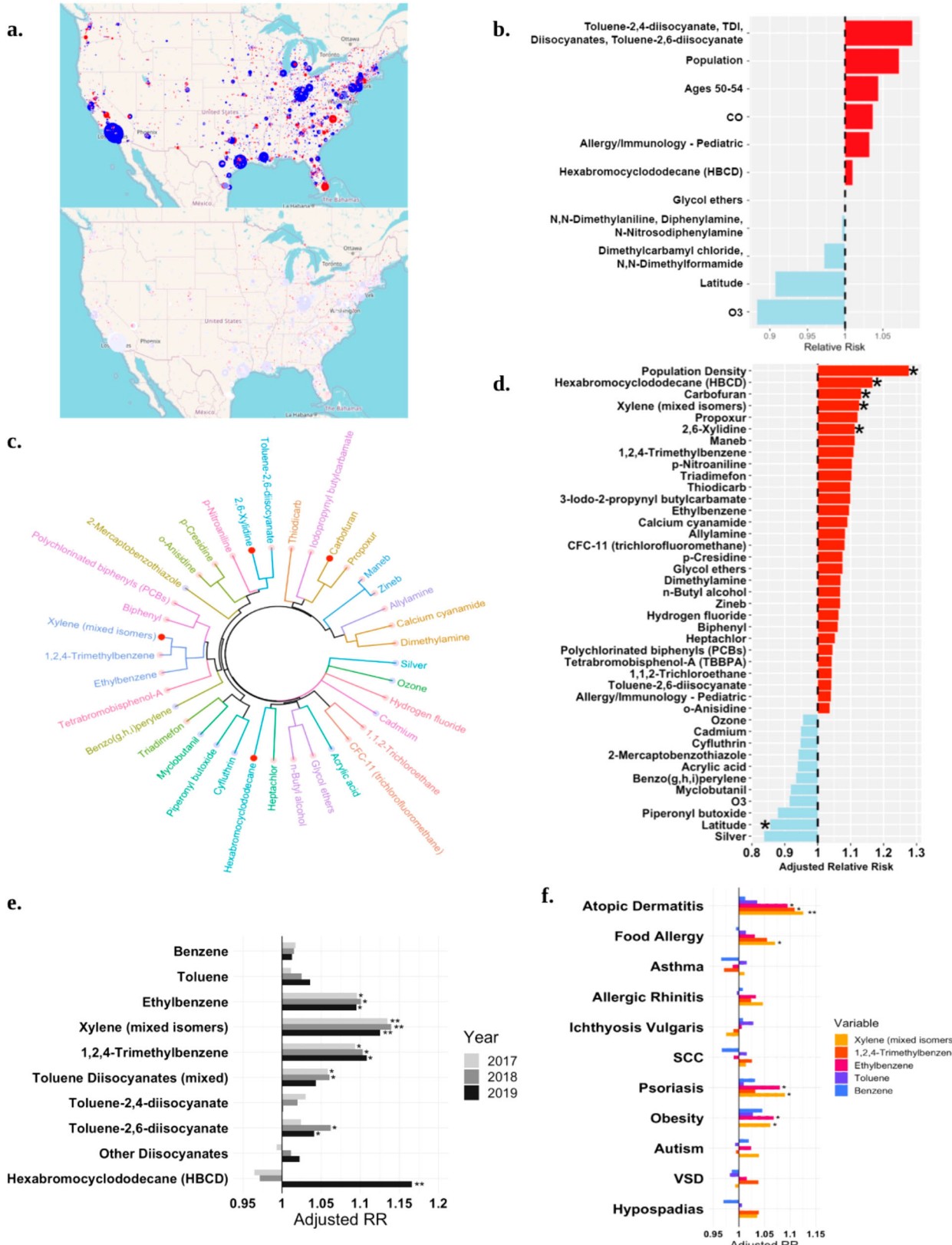

**Fig. 1 | Spatial modeling of AD rates and pollution reveal association with xylene.** **a** Non-spatial lasso Poisson model residual map (upper). Spatially nested random intercept negative binomial model residual map (lower) generated with Tableau public. **b** Relative risk for AD when pollutant variables are clustered by chemical similarity. **c** Dendrogram showing all significant environmental pollutants associated with AD visits clustered by industrial co-release similarity. Red dots indicate positive association; blue indicate negative. Opaque dots indicate chemicals that are still significant after

adjustment for multiple hypothesis testing. **d** All significant exposures associated with AD visits in 2019. Single asterisks signifies significance, double asterisks signifies significance after correction for multiple hypothesis testing. **e** Adjusted relative risk for selected BTEX (benzene, toluene, ethylbenzene, and xylene) compounds and isocyanate-containing compounds and AD for 2017–2019. **f** Adjusted relative risk for BTEX exposure and clinical visits for selected diseases. SCC squamous cell carcinoma, VSD ventricular septal defect, TDI toluene diisocyanate, CO carbon monoxide, O3 ozone.

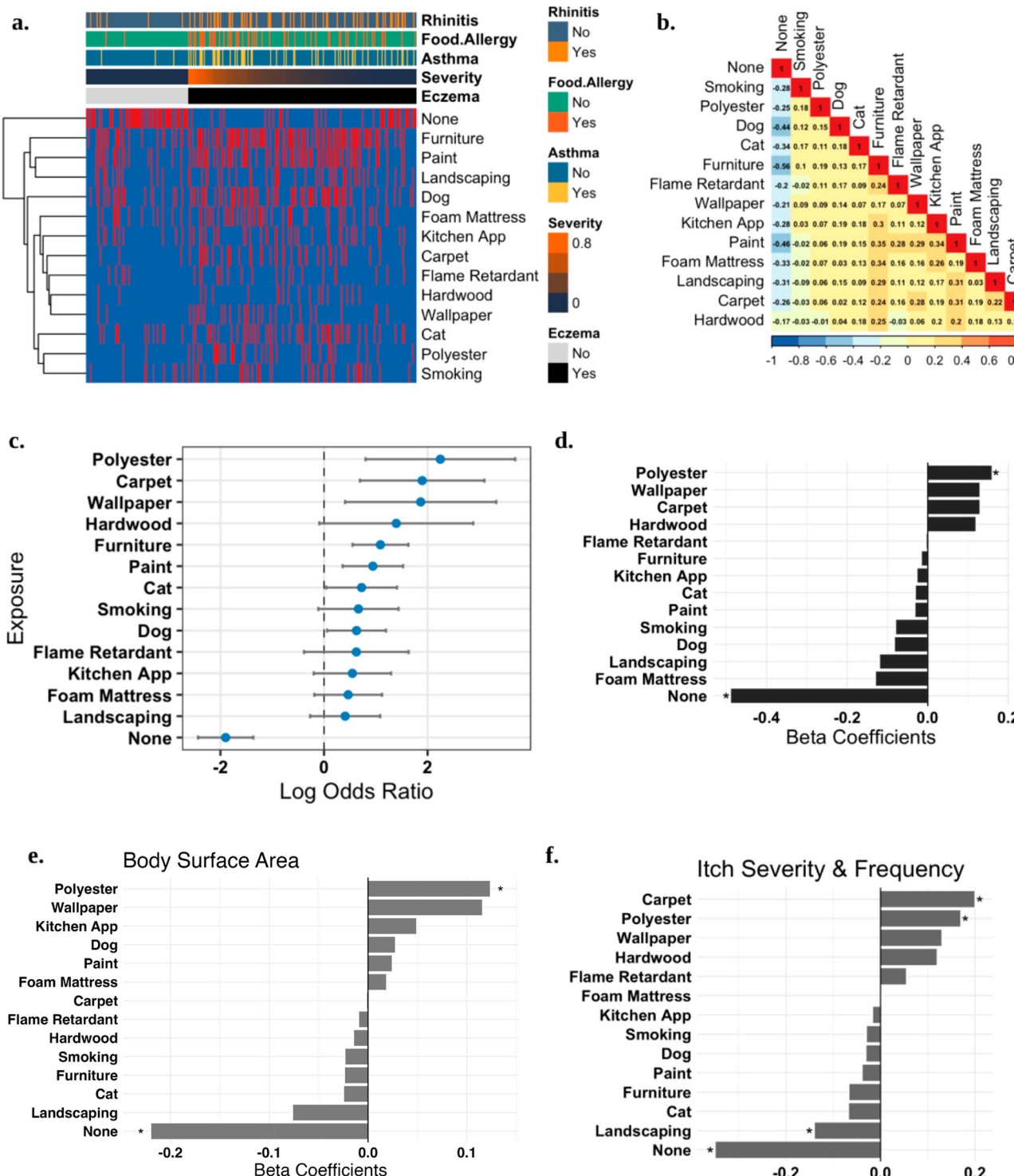

**Fig. 2 | Self-reported childhood BTEX exposure indicates textiles as top in-home risk.** A survey of patients/caregivers and controls (total $N = 316$) on early life (before 3 years of age) exposure to several products that confer BTEX (benzene, toluene, ethylbenzene, and xylene) chemical exposure. **a** Depiction of participant-reported exposures to BTEX products: for survey questions, red is yes, and no is blue. **b** Correlation plot of exposures, VIF (<2.5) indicated no collinearity among variables. **c** Odds ratio and 95% confidence interval for each exposure. **d** Coefficients from a linear model of AD and BTEX exposures. Stars indicate significance ($P < 0.05$). **e** Coefficients from a linear model containing only patients with AD, correlating BTEX exposures and percent BSA affected. **f** Coefficients from a linear model containing only patients with AD, correlating BTEX exposures and a composite metric of itch severity and frequency.

*Staphylococcus epidermidis* (Se), and *Staphylococcus aureus* (Sa) (Supplemental Fig. 3A). After 20 days, Sa persisted on polyester and nylon/spandex, but not on the natural textiles (Fig. 3a). Commensals Rm and Se survived up to 20 days on all fabrics (Fig. 3b, c). Inoculation with a 1:1:1 mixture of all

three species demonstrated commensal dominance on all fabrics by day 20 (Fig. 3d). However, there were significant differences in commensal microbe metabolism by day 3 which persisted to day 20 (Supplemental Fig. 3B, D). *Roseomonas* grown on synthetic textiles and bamboo showed evidence of

BTEX degradation in the dioxin, aminobenzoate, and xylene degradation pathways (Fig. 3e). Compared to cotton, growth of Rm on silk, nylon, and polyester altered the production of alpha-linolenic acids (Fig. 3f). For Se, geraniol degradation and terpenoid backbone biosynthesis were most disrupted by synthetic fabric exposure (Supplemental Fig. 3e).

**Direct BTEX exposure influences microbe metabolism**

Direct exposure to BTEX chemicals indicated that overall metabolism was most similar for ambient air or ethanol control by both nonmetric multi-dimensional scaling (NMDS) and hierarchical clustering (Fig. 4a–d). However, benzene, toluene, trimethylbenzene, xylidine, and xylene all had significant global impacts on Rm (Fig. 4a, b) and Se isolates (Fig. 4c, d). Pathway analysis revealed consistent differences in glycerophospholipid metabolism in Rm and changes in terpenoid backbone biosynthesis in Se (Fig. 4e); each of which is the source of therapeutic benefit in models of AD[7] (Supplemental Fig. 4).

## Discussion

We previously reported that diisocyanate and antimony-containing air pollution are geographically associated with AD visits to healthcare providers[7,14]. In this paper, we enhance our prior model by adding a spatially clustered nested random intercept term to adjust for spatial autocorrelation as well as a more flexible negative binomial structure to account for overdispersion. In this analysis, toluene diisocyanate and antimony compounds were de-emphasized, yet still geographically associated with AD with statistical significance[15] and mechanistic plausibility[6,7,14]. Our spatial model adds to our previous findings by identifying xylene as having multi-year, adjusted significance for associations with AD and representing a potentially broader risk from the related BTEX compounds. However, the spatial model assumes that the random intercept model has the same slope estimates as our fixed effect model. While the spatial model does not address unmeasured confounding, it does offer a new approach to account for the complexity of zip code-level analysis.

Our findings are consistent with the hypothesis that AD has always been predominantly chemically induced, but the modern era has seen a substantial increase in exposure to AD-inducing chemicals. Xylene, (di) isocyanates, and the related BTEX exposure may have been shared between antiquity and today through wildfire smoke, which is an established AD-trigger[16]. However, today, these exposures would more likely come from high-octane gasoline in automobiles and airplanes, catalytic reformation in oil refineries, cigarette smoke, as well as household products like paints, cosmetics, sealants, adhesives, and select textiles[6,7].

Despite numerous patient reports of synthetic textile-induced skin flares, clinical recommendations currently only suggest avoiding textiles with large fibers, such as wool sweaters[13]. This leaves a knowledge gap in both patients' daily decisions for clothing and linens as well as the potential mechanisms of the reported sensitivities. Studies on the growth of bacteria on different textiles have been limited to examining the emergence of bacteria capable of metabolizing these modern chemicals[14,15], the odor of sweat on athletic wear[16], and reducing hospital-acquired Staphylococcal infections through the avoidance of polyester and nylon[17]. The detection of xylene metabolism in the commensals inoculated onto these fabrics suggests the bacteria may breakdown the polymers, however, we cannot exclude residual xylene contamination in the commercial textiles. Our microbial challenge data suggest that synthetic textiles may exacerbate AD symptoms by promoting the growth of *S. aureus* and directing commensal metabolism away from therapeutic pathways. Silver was negatively associated with AD in portions of our analysis and has been used in impregnated clothing as an AD treatment through the inhibition of *S. aureus* colonization[17].

In addition to dysbiotic effects, mounting literature points to BTEX exposure as directly detrimental to host biology. Observational studies have linked BTEX compounds to atopic manifestations, including asthma, milk sensitization, allergic rhinitis, and elevated serum IgE[18]. Mice painted with m-xylene and 1,2,4-trimethylbenzene demonstrated increased expression of

a central allergic cytokine, thymic stromal lymphopoietin (TSLP)[18] and xylene alters methylation of several AD-relevant genes in humans[19]. Similarly, murine inhalation of xylene induces allergic cytokine production and airway constriction consistent with human asthma[20]. In rats, the styrene monomer, a derivative of benzene, increases histamine release in response to dust mite allergens and increases the severity of AD-like lesions at levels 100 times lower than the level currently deemed safe[19]. Air pollution has been proposed as a modulator of dermal aromatic hydrocarbon receptors (AhR), increasing sensitivity to pruritis and AD[21,22]. However, the link between AD-associated chemicals and the thermo-itch receptor TRPA1 may offer a central mechanism for host biology.

In mouse and cell models, TRPA1 activation increases allergic cytokine production such as TSLP and IL-13[23,24], worsens barrier function through impact on tight junctions[25], can be activated by the types of temperature fluctuations known to worsen AD outcomes[5,26], modulates AhR[27], is induced by dust mite exposure[28], and is upregulated on the skin after consumption of a Western diet[29,30]. Both TDI and xylene induce AD-like dermatitis in mice via TRPA1-dependent mechanisms[6,12,13]. Taken together, our work presents a model in which risk factors with population-level associations with AD activate TRPA1, alter endogenous steroid precursors in Gram-positive commensals, and/or suppress ceramide-sphingolipid metabolism in Gram-negative commensals.

Our unifying hypothesis is limited to the pathology on the skin and should not deter from the environmental contributions to barrier erosion and/or dysbiosis in the gut[31]. The co-occurrence of industrial exposures that induce gut dysbiosis (antibiotics, Western diets, farm exposure, etc.) with BTEX exposures will make it difficult to separate the contributions of each. It may be that pathologic reactions to BTEX in the skin may require a background of gut dysbiosis. In addition, various mixtures of chemicals may create varying impacts on skin pathology.

Future work will need to longitudinally measure xylene and isocyanate exposure and correlate it with disease severity in the surrounding populations on an individual level. However, our results indicate that even on a low-resolution level of zip codes, we can find a clear and mechanistically supportable connection between AD and the BTEX toxins. Technological advancements will be needed to perform these types of individual analysis.

Our survey was a preliminary attempt at individual-level analysis but was limited to in-home exposures, and thus cannot directly compare risks from textiles versus exposures from air pollution. Similarly, the survey was administered online and thus could not assure the diagnosis of the respondent, is subject to recall bias, and cannot be demographically matched for the case of controls. Similarly, chemical exposures not contained in the governmental databases cannot be detected by our approach, and thus, untargeted pollution evaluations may provide further insights. Another limitation specific to xylene's potential mechanisms of action is that, unlike diisocyanate[32], xylene was associated with psoriasis in our model. Currently, we cannot distinguish whether the association with psoriasis, such as been previously reported[33] indicates the presence of psoriasis-specific mechanisms or non-specific, accessory pathways for skin inflammation.

However, these limitations are partly mitigated through our approach of combining epidemiologic models with mechanistic testing. Epidemiologic approaches may struggle to identify the specific agents of concern for exposures that consistently contain a fixed group of chemicals. For example, phthalates have been linked to AD, but are used with xylene in the production of polyester[34]. Similarly, particulate matter under 2.5 microns in size (PM2.5) has been linked with AD[35], but is most often sourced from automobile exhaust. The industrial association with BTEX thus complicates assessments of whether phthalates and PM2.5 represent markers of xylene and diisocyanate exposure or are independent risk factors. Screening epidemiologically associated chemicals for their impact on ceramide-family lipids confers a model that connects population-level associations with the molecular pathway most predictive of AD risk[8–10]. Therefore, our findings remain actionable, given that patients and caregivers may benefit from avoiding the implicated fabrics and products. Furthermore, our work

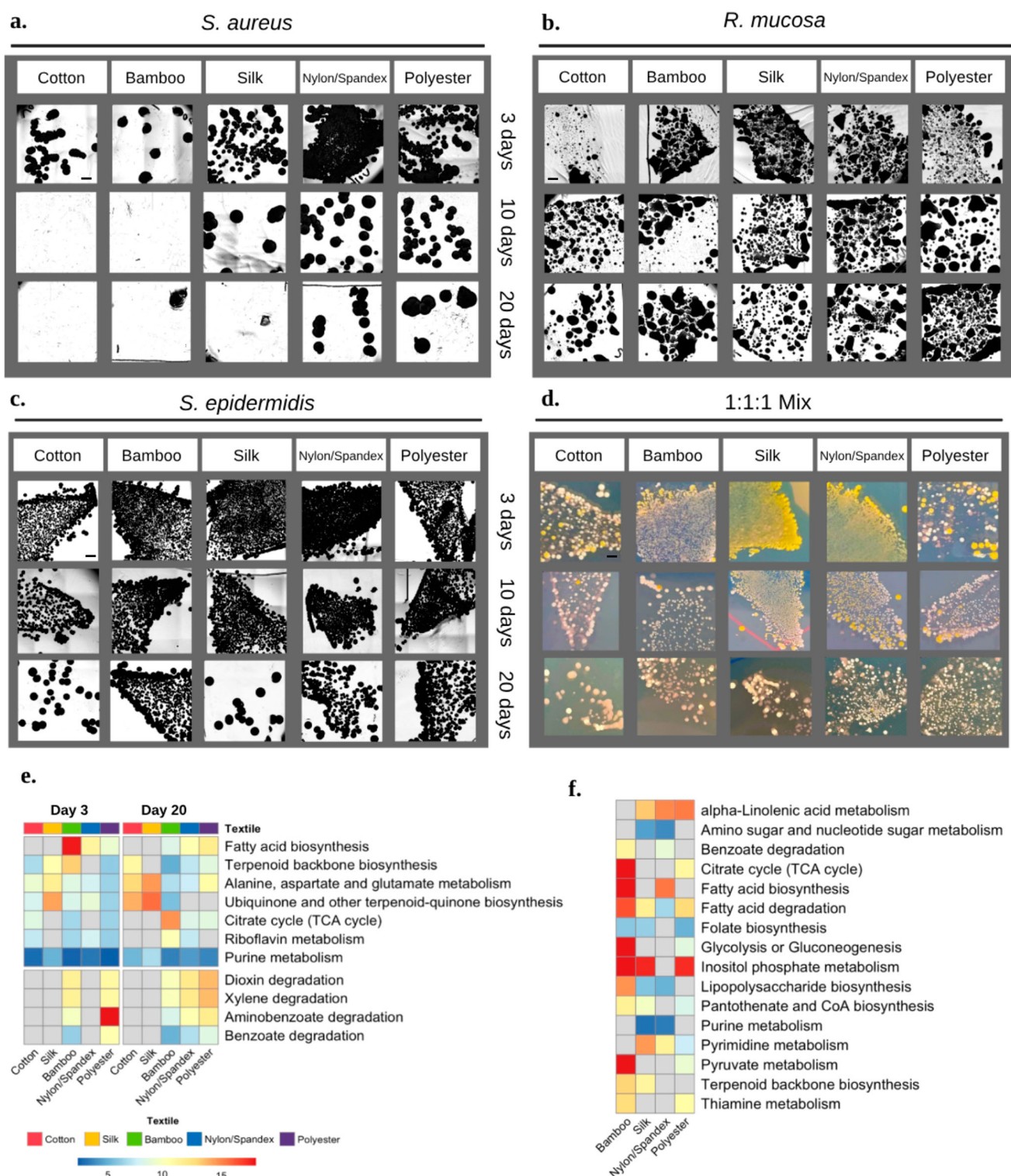

**Fig. 3 | Fabric material impacts commensal survival and metabolism.** Isolates of pathogenic and commensal bacteria were applied to commercially purchased fabric, allowed to sit for 3–20 days, and then were touch-plated onto culture agar for analysis. Bright field microscopy images (all ×1.25 magnification) taken of textile touch plates for **a** *Staphylococcus aureus* **b** *Roseomonas mucosa*, and **c** *Staphylococcus epidermidis* are shown. **d** Images of 1:1:1 competition assay for Sa (Yellow), Se (White), Rm (Pink). For each set of images, scale bar in upper left panel represents 8.2 mm. **e** Heatmap of log-scaled IPS values comparing healthy Rm to Rm grown on different textiles for 3 and 20 days. BTEX (benzene, toluene, ethylbenzene, and xylene)-related pathways were selected in addition to pathways with 6 or more values across samples. **f** Heatmap of IPS values from Day 20 comparing Rm metabolism grown on various textiles to Rm grown on cotton. Pathways with 2 or more values across samples were selected. Results represent three or more independent experiments.

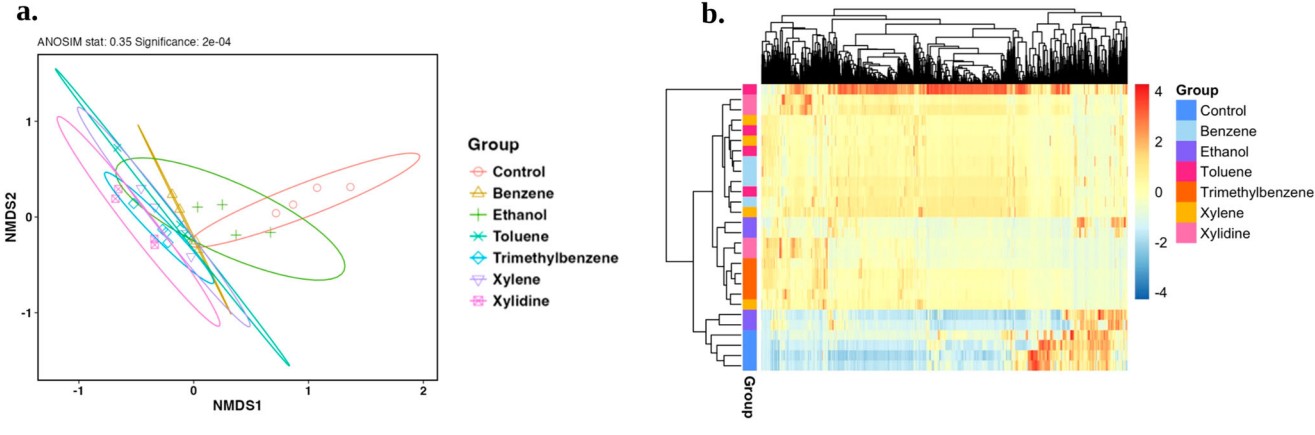

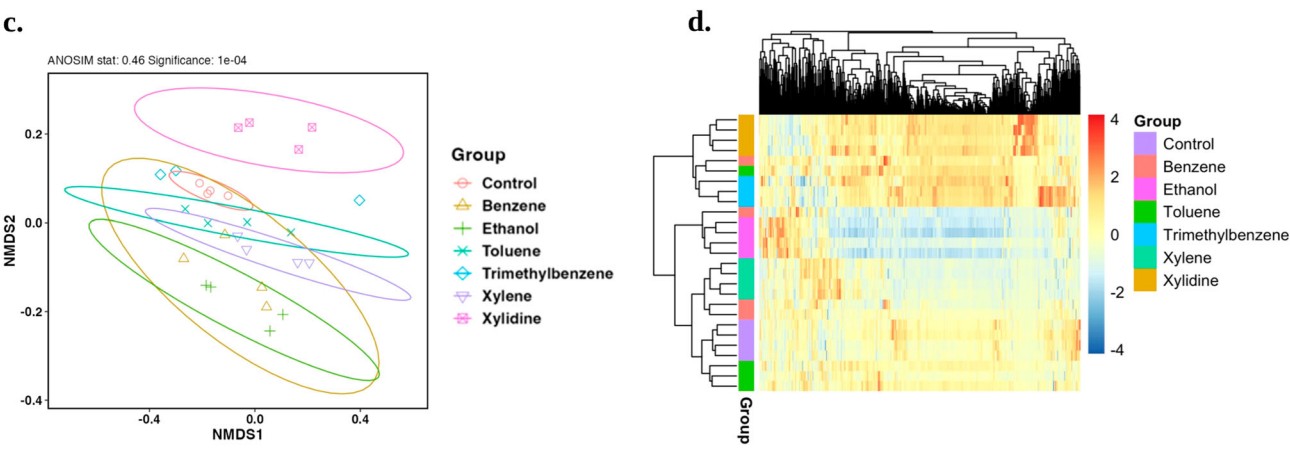

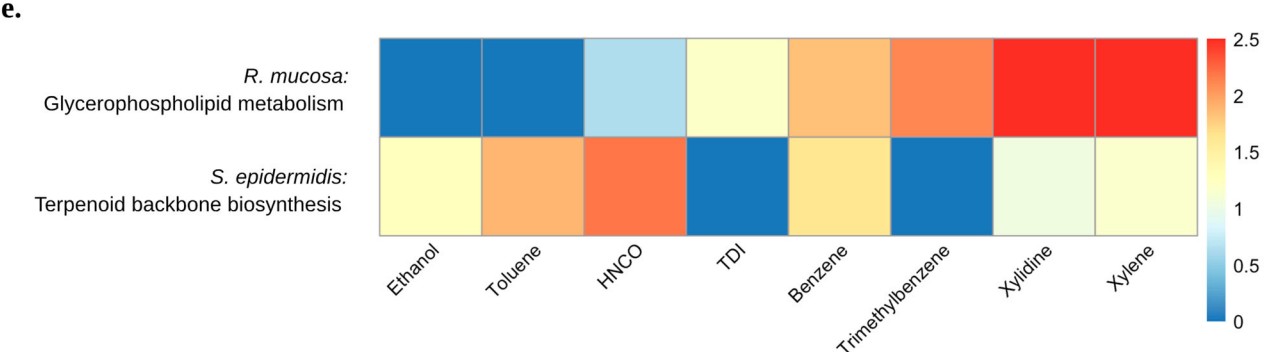

**Fig. 4 | Direct exposure to BTEX chemicals impacts therapeutic metabolic pathways in commensal bacteria.** Commensal and pathogenic bacteria were cultured on agar and then exposed to indicated chemicals for 24 hours prior to analysis. **a** NMDS of overall Rm metabolism and **b** heatmap showing scaled metabolite intensities. **c** NMDS of overall Se metabolism and **d** heatmap showing scaled metabolite intensities **e** Heatmap of log-scaled IPS values for the terpenoid backbone biosynthesis pathway comparing ambient control Se to BTEX (benzene, toluene, ethylbenzene, and xylene) -exposed Se and for the glycerophospholipid metabolism pathway comparing ambient control Rm to BTEX-exposed Rm. **b, d** Dendrograms produced by hierarchical clustering. Results represent three or more independent experiments. TDI toluene diisocyanate.

necessitates a more targeted evaluation of the contributions of environmental injustice to the established racial disparities in allergic diseases[36]. While it would be challenging to eliminate all exposure to these chemicals, our work may guide mitigation policies, as well as the development of filtration systems and alternative products.

## Data availability

Source data for the figures are available as Supplementary Data 1. Metabolomics data will be accessible via MetaboLights identifier MTBLS9879 or will be provided by the authors upon reasonable request. Requests for data from Definitive Health should be directed to the company.

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

## Acknowledgements

The authors would like to thank Mitchell Sumner and Cynthia Gould at RSEI staff, Andreas Bender, Ph.D., and Manas Mahle for technical guidance on methods. We also thank Kelly Barta and Korey Capozza, Ph.D., for helping distribute our patient/caregiver survey. This work was supported by the Intramural Research Program of the National Institute of Allergy and Infectious Diseases (NIAID) and the National Institutes of Health (NIH).

## Author contributions

G.R. designed and conducted all experiments involving fabric and toxin testing on pure cultures and co-cultures, metabolomic experiemnts, assisted in data analysis and visualization, and wrote the manuscript. J.Z. designed and conducted all mathematical models, assisted in data analysis and visualization, and wrote the manuscript. A.A.S. assisted in conducting the patient survey. M.Y. and P.P.C. assisted in selected experiments involving pure and co-culture, metabolomic experiments, and data analysis and visualization. I.A.M. oversaw the project, assisted with select experiments, assisted with data visualization, and wrote the manuscript.

## Funding

## Competing interests

The authors declare no competing interests.
