## [Peer review file · Communications Medicine]

This manuscript has been previously reviewed at another Nature Portfolio journal, prior to review at Nature Communications Medicine.

REVIEWERS' COMMENTS:

Reviewer #1 (Remarks to the Author):

I reviewed this manuscript for a previous publication submission. The authors addressed my major and minor concerns during that manuscript phase. As such, I believe this is suitable for publication here.

Reviewer #4 (Remarks to the Author):

Ratley and colleagues present a well-written study on the possible role of BTEX compounds in fabrics, alongside its effects on *S. aureus* and commensals in microbial culture.

The concerns and extensive questions of previous reviewers have been addressed, especially with regards to pointing out the limitations of the study.

Lines 217-240 sufficiently explain the limitations of the presented hypothesis in the context of air pollution and the effects of the gut microbiome.

While presenting a "unifying hypothesis" of AD risk is quite an ambitious aim, the information presented on the exposure model and microbial metabolism are highly important contributions, while stressing the risks of overabundant synthetics in our environment. Thus, I support publication in Communications Medicine.

Reviewer #5 (Remarks to the Author):

Atopic Dermatitis (AD) is a common chronic inflammatory skin condition whose prevalence has increased 2 – 3 times over the last few decades indicating strong roles played by the environmental factors towards the development of AD. To identify potential pro-AD exposure, Ratley et al used spatial modeling connecting AD prevalence with household exposure to polluting agents. The study therefore addresses a significant unmet need.

Respective datasets (i.e. Definitive Health database and EPA's Toxic Release Inventory) used seem robust. Moran's I has been used for spatial autocorrelation. The analysis is a significant step forward based on the Group's previous publication indicating significant temporospatial and epidemiological association between diisocyanate exposure and the prevalence of AD. The current extended analysis captured further signals connecting exposure to xylene chemicals with AD in addition to several benzene, toluene, ethylbenzene compounds (BTEX).

Retrospective case-control survey indicated childhood exposures to both BTEX-related products and diisocyanate products as significant risk factors for AD. The authors also reported that *Staphylococcus aureus* (an AD-relevant/ pro-AD skin commensal) lived longer on xylene-derived and diisocyanate-derived synthetic textiles compared to natural textiles. Furthermore, their data indicated metabolic shift in the beneficial bacterial species, namely *Roseomonas mucosa* and *Staphylococcus epidermidis*, upon exposure to xylene compounds indicating potential xylene-metabolization, which warrants further investigation. The negative association between thiourea (which can cause skin irritation according to some reports) and AD prevalence might be subjected to future (mechanistic) study.

Collectively, this is a significant spatial-epidemiologic study which utilizes real-world data and is supported by in vitro experiments. Statistical tests seem appropriate. Authors have sufficiently outlined their limitations. Previous studies on gene expression and methylation associated with

xylene exposure (Kim et al Mol Cell Toxicol (2016) 12:15-20. DOI 10.1007/s13273-016-0003-4) indicated that some AD-relevant genes (particularly S100 family genes) may be differentially regulated by xylene exposure which in a way is in line with the rationale of this study.

Regarding authors' responses to comments from reviewer 3 (from previous round of review), Major and minor comments from Reviewer-3 have been well addressed in the revised manuscript and in the rebuttal letter.

Major comments: Regarding the major comments about early life environmental exposures data and recall bias, the investigators mentioned that the survey was administered online and thus could not control multiple factors including individual level exposure, physician-confirmed disease, recall bias, can't follow a conventional case-control study pattern. Limitations have been sufficiently elaborated. The authors have also addressed the comment about missing data handling in their rebuttal letter. The major comment about sample size calculation have been addressed in the revised methods supplements line #91.

Regarding other comments about chemically induced AD signature, availability of birth cohort/exposure data, pollution-atopy connection from historical data (East vs West Germany before unification), the authors' responses seem satisfactory.